# Dietary Intake over a 7-Day Training and Game Period in Female Varsity Rugby Union Players

**DOI:** 10.3390/nu14112281

**Published:** 2022-05-29

**Authors:** Claire Traversa, Danielle L. E. Nyman, Lawrence L. Spriet

**Affiliations:** 1Department of Kinesiology and Physical Education, McGill University, Montreal, QC H2W 1S4, Canada; 2Queens University, Kingston, ON K7L 3N6, Canada; d.nyman@queensu.ca; 3Department of Human Health and Nutritional Sciences, University of Guelph, Guelph, ON N1G 2W1, Canada; lspriet@uoguelph.ca

**Keywords:** nutrition, female athletes, rugby union, energy intake, energy expenditure, macronutrients

## Abstract

This study estimated the daily energy intake (EI) and energy expenditure (TDEE) in female varsity rugby union players during a weekly training/game cycle. Fifteen (nine forwards, six backs) players (20.5 ± 0.4 y, 167.1 ± 1.8 cm, 74.9 ± 2.9 kg) were monitored for a 7-day period (one fitness, two heavy training, one light training, one game, and two recovery days) during their regular season. The average EI throughout the week for all 15 players was 2158 ± 87 kcal. There were no significant differences between days, but the lowest EI (1921 ± 227 kcal) occurred on the mid-week recovery day and the highest on game day (2336 ± 231 kcal). The average TDEE was 2286 ± 168 kcal (~6% > EI). The mean energy availability (EA) over the 7-day period was 31.1 ± 3.6 kcal/kg FFM/day for the group. Of the players, 14% were in the optimal EA range (>45 kcal/kg FFM/day); 34% were in the moderate range (≥30–45 kcal/kg FFM/day); and 52% had a poor EA of <30 kcal/kg FFM/day. Carbohydrate (3.38 ± 0.36 g/kg/day, 45% of EI); fat (1.27 ± 0.12 g/kg/day, 37% of EI); and protein (1.38 ± 0.12 g/kg/day, 18% of EI) consumption remained similar throughout the week (*p* > 0.05). The players consumed 6% less energy than they expended, providing poor to moderate EA; therefore, daily carbohydrate intake recommendations were not met.

## 1. Introduction

Rugby union is an intermittent stop-and-go sport involving moments of high-intensity and high-energy demand, such as sprinting or tackling, which are interspersed with lower intensity periods, such as jogging and walking [1]. Elite level players require high levels of anaerobic and aerobic energy provision, as well as optimal muscular endurance, strength and power, agility, and speed to keep up with the physical demands of the sport [1]. Typically, the forward positions are more involved in grappling and physical collisions, are greater in stature and body mass, and carry a higher percentage of body fat [2]. The back positions are predominately involved in sprinting and running [3]. Following participation in a match, significant muscle damage and fatigue has been recorded, lasting up to 48 h before returning to baseline levels [1]. This has also been correlated with a high number of musculoskeletal injuries [3]. Recovery processes and injury prevention can be improved with proper physical training and adequate nutrition.

Though we may regard athletes as nutritional role models, many do not achieve optimal nutritional practices. Zuniga et al. [4] revealed that the majority of university level programs do not have access to a nutritionist. The literature has frequently reported that female athletes do not meet their energy intake (EI) goals given their high daily energy expenditure (TDEE) [5]. Manore [5] speculated that in females, this issue could stem from trying to maintain body image or body weight goals. In any case, inducing the restriction of EI for weight loss or body composition changes can directly impede the development of an athlete and pose a threat to their overall health [6]. 

Carbohydrates (CHO) are the most important macronutrient to consume for individuals engaging in high-intensity stop-and-go sports as they help to restore liver and muscle glycogen reserves following a bout of exercise, where near exhaustion of these stores is associated with fatigue, impaired work rate, and loss of concentration [7,8,9]. Protein is also critical to the athlete’s diet due to its involvement in muscle protein synthesis, which can remain upregulated for ≥24 h following a bout of resistance exercise [8,9,10]. Many female athletes may be unaware that the consumption of protein following a bout of resistance exercise is responsible for the preservation of lean tissue, in addition to the building, repair, and regeneration of muscle tissue [7]. The importance of fat intake for female athletes, or athletes in general, is less common than CHO and protein in the literature.

The International Olympic Committee (IOC) released a consensus statement outlining how low energy availability (LEA) in female athletes can impair their metabolic rate, menstrual function, bone health, immunity, protein synthesis, and mental and cardiovascular health in the short-term [11,12]. This results from a mismatch between daily EI and TDEE such that there is insufficient energy for normal bodily functions once the energy expenditure required for exercise (EEE) is accounted for [11,12]. When LEA occurs in a chronic state (resulting from the body eventually adapting to operating with inadequate EA), this becomes relative energy deficiency syndrome (RED-S) [13]. Sports nutrition is a growing subject of study to continuously push the level at which athletes perform; however, there appear to be gaps in what we already know and how we are educating our athletes with this information. To support the development and success of female athletes, more female-specific research is needed surrounding nutrition efforts. 

This study was designed to gain knowledge of the existing dietary patterns of female varsity rugby union players representing a diverse population of body types and energy needs. This study measured energy and macronutrient intake and estimated the energy needs of female varsity rugby union players during a typical 7-day monitoring period throughout their regular competitive season. This information was used to evaluate the average energy availability (EA) experienced by female rugby players during various load days (practice, game, recovery).

## 2. Materials and Methods

### 2.1. Participants

A total of 15 female varsity rugby union players (9 forwards, 6 backs) volunteered to participate in the study. Athletes were excluded if they were considered injured at the time of volunteering or if they had a diagnosed condition preventing them from fully participating in practices/games. All participants were provided with the study protocol upon recruitment. Those who agreed to participate were required to provide written consent. Ethics approval for this study was received from the Research Ethics Board of the University of Guelph. Characteristics of the participants are outlined in Table 1.

### 2.2. Experimental Design

The participants were randomized into 3 groups of 5, each of which included both forwards and backs. The participants were instructed to record everything they consumed for 7 full days. Two participants asked for their assigned week to be delayed due to pressure from schoolwork and were then pushed back to an additional 4th week to improve compliance. All meetings took place with the lead researcher. At the commencement of their participation week the athletes were met for an introductory meeting following their Sunday morning recovery session to begin recording the next day (Monday). All participants underwent baseline anthropometric measurements. Height was determined using a stadiometer (Seca, Chino, CA, USA) and weight using a traditional digital scale (Seca, Chino, CA, USA). Each participant was provided with a binder containing the consent form, an outline of the study protocol including tips, suggestions, and serving size examples, and blank diet record sheets. Each participant was provided with a Starfruit food scale (Longueuil, QC, Canada) accurate to ± 0.1 g, a set of measuring cups (1/8 cup–1 cup), and a set of measuring spoons (1/4 teaspoon–1 tablespoon). The participants were introduced to the “MyFitnessPal” application (Under Armour, Version 22.8.0, San Francisco, CA, USA, 2019); they created an anonymous account for the researcher to follow throughout the study and were instructed on how to properly enter a food item, and any other requirements for their daily journal (i.e., exercise logs). Those who did not wish to use the app were given blank record sheets with a sample log to record via pen and paper. All participants were shown how to properly use a food scale and were provided insight on how to record a meal ordered at a restaurant or a meal not made by themselves. Following the commencement of the study, daily meetings were held with the lead researcher to ensure compliance and answer any questions the participants had. In some cases, the participants had taken photos of their meals, and had the researcher walk them through the best way to enter that meal.

### 2.3. Anthropometry Measurements

Body composition data (player lean body mass (kg)) were measured in our laboratory as part of a related study [14] via bioelectrical impedance analysis (BIA) (Bodystat 1500; Bodystat Ltd., British Isles) and silver/sliver-chloride electrodes (SilveRest; Nisha Medical Technologies, Buffalo, NY, USA). Players were instructed to consume 500 mL of water 1 h before BIA measurements to control for hydration status (Table 1).

### 2.4. Weekly Training Schedule

Each testing week took place over 7 days to include 4 practice days, 2 recovery days, and a game day. The women’s varsity rugby team typically follows the same weekly schedule (Table 2).

The theme of the practice appears in quotations, where “Fitness” implies the practice had a low amount of contact and a high amount of aerobic running drills. “Heavy” practice took place on Tuesdays and Thursdays where the players took part in a full-contact scrimmage, usually taking place during the second hour of practice and preceded by high contact drills, position-specific skill training, and a review of plays. “Run-Through” was a light, no contact practice that took place the day before a game to review both offensive and defensive tactics. From this schedule, the only days to overlap in routine were Tuesdays and Thursdays. Therefore, there were 6 different day types, labelled throughout this study:**Monday:** “Fitness”**Tuesday and Thursday:** “Heavy”**Wednesday:** “Mid-Week Recovery”**Friday:** “Light”**Saturday:** “Game”**Sunday:** “Post-Game Recovery”

We decided to examine each of the two recovery days as separate day types since one occurred during the week when the players still attended classes, seminars, study hall, etc., and the second took place on a weekend day when the players were off for the full day.

### 2.5. Energy Expenditure Estimates

To calculate the total daily energy expenditure (TDEE) requires the following components: basal metabolic rate (BMR); thermic effect of food (TEF); and energy expenditure of exercise (EEE).
**TDEE = BMR + TEF + TDEE**

An estimate of each player’s BMR was achieved using their height (cm); body mass (BM, kg); and age (years). The Harris—Benedict equation [15] as access to indirect calorimetry was not available: **BMR = 655.1 + 9.563 (BM) + 1.850 (height) − 4.676 (age)**

The TEF was estimated from the ESHA software-derived EI for each participant. ESHA estimates this value as 10% of total food consumption (kcal):**TEF = (kcal_total_ × 1.1) − kcal_total_**

Multiple equations were used to produce the most accurate estimate of EEE. EEE estimates for rugby practices were acquired from data that was obtained during a related study in our laboratory [14]. These data used a 10 Hz GPS tracking unit (PlayerTek Pod; Catapult Sports, Chicago, IL, USA) that the players wore secured between the scapulae in a sports bra for the duration of practice (repeated for 3 practices across one week, each player). Practices took place on a natural grass pitch (70 × 144 m) during the months of September and October, corresponding to their regular season. The Catapult software automatically produced an EEE value that was based on the duration of exercise and geographical distance covered. The validity of the GPS-derived energy expenditure estimates has not been confirmed in female athletes but has provided reasonable estimates for contact team sports in men [16]. These data were used to derive the average values for forwards and backs of energy expenditure per minute, then used to predict the total energy expenditure during practices and games by multiplying the average value by the duration of the practice or the number of minutes played in a game.
**Forwards = EEE_rugby_ = 6.27 (kcal/min) × (minutes participated)**
**Backs = EEE_rugby_ = 5.57 (kcal/min) × (minutes participated)**

On the days where players had strength training workouts (referred to as “Lift”) in addition to practice, a physical activity factor of 0.05 (a standard value for moderate intensity strength training) was multiplied by their BMR and the number of participation minutes that were recorded in their log for the workout on that day. This value was then combined with the EEE_rugby_ value to produce a TDEE value.
**EEE_lift_ = (0.05)(BMR)(minutes participated)**
**TDEE = EEE_rugby_ + EEE_lift_**

### 2.6. Energy Availability

EI and EEE (kcal) over the 7-day monitoring period were used to calculate EA, where EA = (EI − EEE)/kg FFM, and FFM is fat free mass [17]. EA was categorized as optimal (>45 kcal/kg FFM/day); moderate (30–45 kcal/kg FFM/day); or poor (<30 kcal/kg FFM/day) [17,18]. A hypothetical calculation was conducted to add 10% EI to all data (to compensate for possible under-reporting) to evaluate how this would affect the EA values throughout the week.

### 2.7. ESHA and Microsoft Excel Analysis

Participants’ 7-day dietary records were manually entered into an ESHA Food Processor (ESHA Processor Nutrition Analysis Software, Salem, MA, USA). The lead investigator was trained on the ESHA Food Processor software. From this software, the total energy and macronutrient intakes were averaged for each of the 6 day-types in Microsoft Excel Version 16.6 (Microsoft, Washington, WA, USA, 2022), and further dissected by position (forwards and backs). The data was used to determine the average relative amounts (g/kg) that were consumed for each macronutrient, for each of the 6 day-types.

### 2.8. Statistics

All statistical analyses were completed using SPSS Statistics, Version 27 (IBM, New York, NY, USA, 2020). A two-tailed paired sample t-test was used to compare average EI and average TDEE (*p* < 0.05). EA was compared across the 6 day-types using a multi-factor ANOVA for the entire sample. When the significance was determined (*p* < 0.05), a Tukey’s multiple comparison *post hoc* test was used to determine the significance between the groups.

## 3. Results

### 3.1. Total Energy Expenditure and Total Dietary Intake Results

The average EI over 7 days for all 15 players was 2158 ± 87 kcal (Table 3). There were no significant differences between days, but the lowest EI (1921 ± 227 kcal) occurred on the mid-week recovery day and the highest on game day (2336 ± 231 kcal) (Table 4).

The average estimated TDEE over the 7 days was 2286 ± 169 kcal (Table 3). Therefore, players consumed an average of 6% less energy than their estimated TDEE. TDEE differed significantly between all days (*p* < 0.05), except for the two recovery days (*p* > 0.05) (Table 4). Significant differences (*p* < 0.05) in EI vs. TDEE were seen on the heavy training day (TDEE > EI, *p* < 0.01) and the post-game recovery day (EI > TDEE, *p* = 0.029) (Table 4). 

The forwards had an average EI of 2144 ± 208 kcal vs. their average TDEE of 2397 ± 38 kcal, a net −11% difference (Figure 1, Table 5). When evaluating by day type, the forwards appeared to under-consume on the heavier load days (fitness, heavy, and game) and then over-consume on the lighter load days (mid-week recovery, light, and post-game recovery) (Table 5). When EI vs. day was compared for the forward group, no significant differences were found. When TDEE vs. day type was compared for the forward group, all days were significantly different (*p* < 0.01), except for the two recovery days, and the light vs. game day (Table 4). 

The backs had a weekly average EI of 2049 ± 116 kcal vs. their average TDEE of 2118 ± 20 kcal, a net -4% difference (Figure 1, Table 5). A one-way ANOVA revealed a significant difference between the TDEE of forwards and backs (forwards > backs, *p* < 0.01), but not the EI (Figure 1). The backs under-consumed on fitness, heavy, and light days, whereas over-consumption was seen on the two recovery days and game day (Table 5). Comparing EI vs. day type for the backs group did not detect any significant differences. When TDEE vs. day type was compared for the backs group, all days were significantly different (*p* < 0.01), except for the two recovery days and the fitness vs. game day (Table 4).

### 3.2. Energy Availability

The mean EA over the 7-day period was 29.0 ± 3.7 (kcal/kg/FFM/day) for the forwards, 33.3 ± 3.5 for the backs, and 31.1 ± 3.6 for the group overall. There were no significant differences in EA when game day was compared to both the fitness and light training days (*p* > 0.05) in either position group (Figure 2). The EA range was divided into three categories: optimal EA (>45 kcal/kgFFM/day); moderate EA (≥30–45 kcal/kg/FFM/day); and poor EA (<30 kcal/kgFFM/day). Overall, an average of 14% of players were in the optimal range throughout the week, 34% were in the moderate range, and 52% had a poor EA of <30 kcal/kgFFM/day (Figure 3). Even after adding 10% onto the EI values for each day (to potentially account for EI under-reporting), the new weekly EA averages became: 26% of players in the optimal range, 34% in the moderate range, and 40% with a poor EA of <30 kcal/kgFFM/day, despite making TDEE and EI match.

The forwards’ EA was significantly less on heavy days compared to all other days, (fitness *p* = 0.017; mid-week recovery *p* < 0.01; light *p* = 0.010; game *p* = 0.011; post-game recovery *p* < 0.01) (Figure 2). Significantly higher EA was found on the post-game recovery day, compared to the fitness day (*p* = 0.05) and the heavy day (*p* < 0.01) for the backs (Figure 2).

### 3.3. Average Macronutrient Intake/Distribution Results

The average macronutrient distribution for the varsity female rugby player was: CHO (3.38 ± 0.4 g/kg/day, 57% of EI); fat (1.27 ± 0.1 g/kg/day, 20% of EI); and protein (1.38 ± 0.1 g/kg/day, 23% of EI) (Table 3). This distribution remained similar throughout the week (*p* > 0.05) and CHO intake was more than double the intake of both fat and protein (Table 4). When compared to the recommended daily intake amounts (Table 3) for athletes, it was found that 12% of participants consistently met CHO recommendations; 50% of participants consistently met fat recommendations; and 55% of participants consistently met protein recommendations.

## 4. Discussion

### 4.1. Energy Intake vs. Energy Expenditure Estimates

Overall, this study reported the average EI to be 6% less than the average TDEE. A similar result was reported by Vermeulen and colleagues [19] in varsity level female ice hockey players. Several measures were put in place to ensure compliance during diet recording (i.e., using a mobile application to record in real time, conducting daily meetings with the participants to monitor progress, allowing participants to send photos of their meals when eating out). It is well documented that females often under-report during dietary record procedures with literature values ranging from 10–30% of participants being considered “under-reporters” [20]. Similarly, self-reported diet-recording compliance has been seen to drop significantly after 4 days [13], and this study required 7-days of diet recording. There are also few tools available to properly distinguish between true under-reporting and dieting [21]. However, females are often more likely to struggle with eating habits and body image than their male counterparts [5,22]. It should also be mentioned that there is a possibility that players changed some dietary habits during the week of diet recording, being mindful that their choices would be reviewed.

We estimated TDEE using a series of equations as a best attempt to find the most accurate representation of TDEE in female rugby union players. Few studies have used a GPS unit to measure the distance and speeds that are reached during practice and games for female rugby union players [14,23,24,25]. This study is unique in using these data to produce an estimate for energy expenditure per minute. However, this value does not incorporate heart rate or absorption/delivery of contact (tackling) which would have improved our confidence in the accuracy of the EEE_rugby_ value. The TDEE equations also do not factor in non-purposeful exercise throughout the day (i.e., walking to class, climbing stairs, etc.) [13], nor was the estimation for BMR taken via indirect calorimetry or a method of similar validity. By considering that both EI and EE estimations have margins of error, we do think that our conclusion of female varsity rugby union players consuming an average of 6% less energy than they expend to be as accurate as possible. Pre- and post-season body masses would have been the best opportunity to confirm this conclusion but these were not taken during this study. A similar study looking at female NCAA Division I basketball and softball players and comparing their EI and body composition at the beginning and end of season saw no changes in body composition, but it did see an increase in EI at the end of the season, compared to the beginning [26]. This may indicate that varsity athletes go through weeks of adequate EI and weeks of inadequate EI (or even alternate days) which is why dramatic body composition changes are not seen. Alternatively, by consistently consuming less energy than the body needs, the body adapts by limiting the energy it provides for other physiological functions (bone health, immunity, menstrual cycle, etc.) and not necessarily body composition [27]. Mountjoy and colleagues [27] determined that the average energy consumption for female athletes should be between 30–45 kcal/kg to support all of the bodies’ functions and avoid LEA status. In the current study, the participants averaged 31.1 kcal/kgFFM/day throughout the week, with some days being much higher than others. The diagnosis of RED-S may explain why athletes would not experience any differences in body mass throughout the season, although a non-athletic control group would also be useful in confirming this—which merits further investigation in female varsity athletes. Furthermore, the phenomenon of fluctuating between adequate and inadequate EA in a short season such as rugby may be enough to maintain performance parameters and body composition.

### 4.2. Energy Availability

Reviewing the EA of all study participants revealed that 86% of players did not achieve optimal EA during the 7 days of monitoring. This is similar to the results that were reported by Moss and colleagues in professional female soccer players [17]. The proportion of players meeting this goal was highest on the post-game recovery day. In a similar manner, the proportion of players in the poor EA range (<30 kcal/kgFFM/day) was highest on the heavy training load days. This phenomenon where an increase in EEE is not accompanied by an increase in EI has been commonly recorded in female athletes (more so than their male counterparts) [27,28]. In the varsity athlete population specifically, it is difficult to determine whether this discrepancy stems from a lack of knowledge, a lack of preparation, or the inability to prepare for ever-changing schedules as a student-athlete. In addition, previous studies in the general college student population [29] have highlighted appetite suppression under stress. One might consider varsity athletes as having a double burden of stress (from both school and sports) which could explain the potential for under-eating due to appetite suppression. 

Though additional characteristics may be needed to confirm the diagnosis of RED-S, we can confirm that 86% of players performed during LEA (<30 kcal/kgFFM/day) using the methods carried out in this study (EI vs. EEE vs. body composition) [13]. Though there is certainly individual variability on how the effects of this may manifest, previous reports have shown that even short-term LEA can have unfavorable effects on muscle protein synthesis, muscle glycogen stores, and serum hormone levels [30]. These effects may directly impact muscle performance potential and increase injury potential due to continual decreases in muscle mass size and strength that result from declines in muscle protein synthesis and estrogen/progesterone rates [30]. The inability to properly recover from full contact practices/games that is caused by consistently depleted glycogen stores is also presented with LEA—specifically, surrounding CHO availability deficiency. Unfortunately, standardized performance tests were not implemented in this study to evaluate whether any performance declines were seen throughout the week. In an intermittent sport, it is also undetermined whether increasing EI and subsequently increasing EA would improve performance during the short-term competitive season. This could be an avenue for future research in rugby union players.

### 4.3. Macronutrient Intake

#### 4.3.1. Carbohydrates

The average CHO intake was determined to be 3.4 g/kg of CHO or 57% of the diet of female varsity rugby union players (Table 3). This amount of CHO consumption may be acceptable for the general population, however, for athletes undergoing 1–3 h of intense physical exercise a day, the recommendation is between 6–10 g/kg to support their energy needs during workouts, as well as during the recovery period [8,9]. CHO play an important role in providing fuel for the brain and central nervous system, as well as anaerobic and oxidative muscular functions [8,31]. In sports, where decision making is involved, CHO as fuel for the brain is especially important. The brain alone is responsible for approximately 25% of all glucose consumption, relying solely on CHO [8]. Our results were similar to results that were reported in female Australian rules football players (*n* = 30) aged 18–35 (weight: 64.5 kg ± 8.0; height: 168.2 cm ± 7.6) who, after a 24 h Dietary Assessment were found to consume an average of 3.0 g/kg CHO per day [32]. A similar study carried out in varsity level female ice hockey players found that their athletes consumed a daily average of 4.6 g/kg CHO [19]. While this is higher than what our participants were able to achieve, it still falls short of recommendations for athletes of this caliber. Moderate-exercise athletes should be aiming for 5–7 g/kg/day and moderate-to-intense exercise athletes should be aiming for 6–10 g/kg/day [30]; however, a systematic review of dietary intakes conducted in professional and semi-professional team sports also found both EI and CHO to be the consistent shortcomings across several team sport studies [33]. It has been noted that 6–10 g/kg of CHO intake can be difficult for females to achieve since they more commonly consume unprocessed, low energy dense CHO (i.e., vegetables, whole fruits) [31]. These types of CHO are recommended to fulfill the needs of a healthy diet that is nutrient dense and rich in fiber content, however, the timing of their consumption in relation to timing of exercise can be a key factor in CHO availability during energy expenditure [7,8]. Excessive consumption of these nutrient sparse CHO while training can result in a low energy diet and fewer calories consumed which may explain the inadequate EI noted in the previous section [12,34]. Another element that may have been helpful in our methodology would have been a nutrition questionnaire to find out how many athletes were aware that these are the benchmark values they should be aiming for in their daily nutrition. On the other hand, knowing that the majority of these nutrient suggestions have been derived from data in male-only or endurance sport studies, and considering females seem to be struggling to meet them, do we need to revisit our nutrient recommendations for female varsity athletes? 

#### 4.3.2. Protein and Fat

Average protein intake was determined to be 1.4 g/kg or 23% of the diet of female varsity rugby union players (Table 3). Current research suggests that to support muscle repair, remodeling, metabolic adaptations, and possibly muscle building, athletes should be consuming between 1.2–2.0 g/kg body weight of protein per day [8]. Therefore, the sample in this study is within the recommended daily intake for elite athletes of 1.2–2.0 g/kg/day [8]. A limitation around protein intake was dose timing. Due to the nature of the MyFitnessPal application that was used to record EI, the time of entry was not visible to the researcher. Exploring protein timing may be beneficial to athletic performance, which would merit future investigation in this population.

Average fat intake was determined to be 1.2 g/kg or 20% of the female varsity rugby players’ diet (Table 3). This is well within the recommended daily intake for elite athletes of 0.5–1.5 g/kg/day [8]. These levels do not merit concern for an inadequate amount of fat intake which is important for several biological functions [7]. Possible future objectives may be to quantify the amount of omega-3 fatty acids, eicosapentaenoic acid (EPA) and docosahexaenoic acid (DHA) in the diet of this population. EPA and DHA are significant in the attenuation of inflammation, particularly in the brain, which in a contact sport can help prevent or heal concussions, as well as other injuries [7]. Conclusively, both the protein and fat intake levels of the participants were adequate for this caliber of athlete.

## 5. Conclusions

Rugby is a highly demanding intermittent sport. Female varsity rugby players consumed an average of ~6% less than their estimated TDEE over 7 days of diet during their regular competitive season. The players had an average EA of 31.1 kcal/kgFFM/day, putting them at risk of short-term LEA. The optimal recommended intake of female athletes of this caliber is 45+ kcal/kgFFM/day, which over 80% of players did not meet. The average daily macronutrient intakes were determined to be 3.4 g/kg of CHO, 1.2 g/kg of fat, and 1.4 g/kg of protein. CHO intake was most concerning, as recommendations indicate that female athletes of this activity level should be consuming between 6–10 g/kg/day. Fat and protein intakes were both within the recommended intake levels. In the future, it would be useful to determine more accurate methods of estimating EEE in female athletes. The ongoing concern of diagnosing and treating RED-S could additionally help to provide insight on how to best educate athletes of this population on their nutrition habits.

## Figures and Tables

**Figure 1 nutrients-14-02281-f001:**
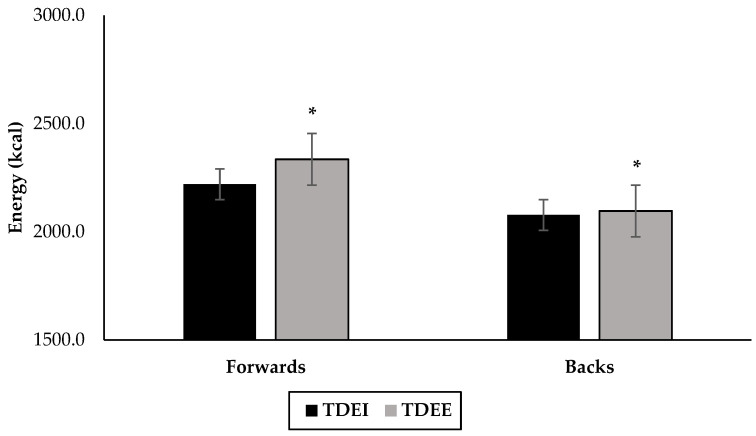
Average weekly values for total daily energy intake (TDEI) and total daily energy expenditure (TDEE) in both forward and back positions (*n* = 15, 9 forwards, 6 backs). * Significant difference from TDEI, same position (*p* < 0.01).

**Figure 2 nutrients-14-02281-f002:**
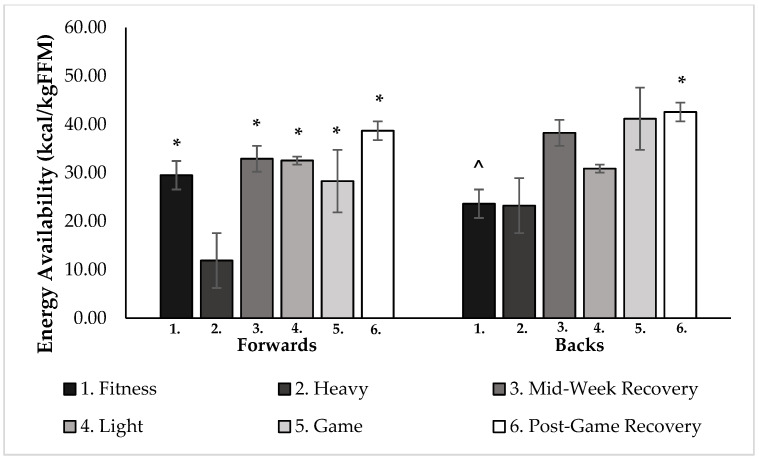
Average daily EA values for 6 different day loads for forward and back positions (*n* = 15, 9 forwards, 6 backs). * Significantly different from heavy day (*p* < 0.05). ^ Significantly different from post-game recovery day (*p* < 0.05).

**Figure 3 nutrients-14-02281-f003:**
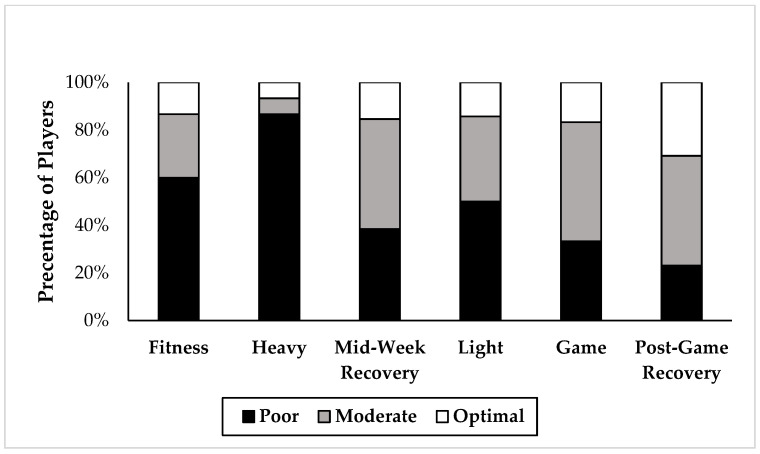
The relative percentage of players in optimal (white bars), moderate (grey bars), and poor (black bars) EA for 6 different training load days where optimal EA is >45 kcal/kgFFM/day, moderate EA is ≥ 30–45 kcal/kg/FFM/day and poor EA is <30 kcal/kgFFM/day (*n* = 15).

**Table 1 nutrients-14-02281-t001:** Average anthropometric characteristics and basal metabolic rate (BMR) of the participants.

Characteristic	Average	Range
Age (year)	20.5 ± 0.4	18.0–23.0
Body mass (kg)	74.9 ± 2.9	55.3–88.9
Lean body mass (kg)	55.9 ± 1.6	41.2–66.9
Height (cm)	167.1 ± 1.8	162.5–180.3
BMR (kcal/day)	1609.3 ± 25.6	1400.7–1717.0

Means ± SEM, *n* = 15. BMR, basal metabolic rate.

**Table 2 nutrients-14-02281-t002:** Women’s varsity rugby team weekly schedule.

Day	Activity	Time
Monday	“Fitness” practice	17:00–19:00
Tuesday	Lift “Heavy” practice	07:00–08:00 17:00–19:00
Wednesday	Film (mid-week recovery)	18:00–19:00
Thursday	Lift “Heavy” practice	07:00–08:00 17:00–19:00
Friday	“Run-Through” practice Team dinner	17:00–18:00 19:00–21:00
Saturday	Warm-up Game	11:45–12:45 13:00–15:00
Sunday	Post-game recovery	Full Day

**Table 3 nutrients-14-02281-t003:** Average weekly energy expenditure and energy and macronutrient intakes in comparison to recommendations for all participants.

Measure	Recommendation	Average	Range	Athletes Meeting Recommendations
TDEE (kcal)		2286 ± 169		
EI (kcal)		2158 ± 87	613–4423	
(kcal/kg)		38.6 ± 2	9–76	
CHO (g) (g/kg)		246.9 ± 12.2	34–659	12%
	6–8 [8,17]	3.4 ± 0.2	<1–8	
Fat (g)		91.0 ± 4.6	23–246	50%
(g/kg)	1.2 [7,8]	1.2 ± 0.1	0.3–3.0	
Protein (g)		99.6 ± 4.6	13–238	55%
(g/kg)	1.2–2.0 [8]	1.4 ± 0.1	0.2–3.2	

Means ± SEM, *n* = 15. TDEE, total daily energy expenditure; EI, energy intake; CHO, carbohydrate.

**Table 4 nutrients-14-02281-t004:** Average daily energy expenditure and energy and macronutrient intake over 6 different load days for team.

Measure	Fitness	Heavy	Mid-Week Recovery	Light	Game	Post-Game Recovery
TDEE (kcal)	2520 ± 49	3018 ± 54 *	1782 ± 33 ^	2149 ± 43	2295 ± 42	1802 ± 35 * ^
EI (kcal)	2236 ± 216	2101 ± 185	1921 ± 227	2138 ± 243	2335 ± 231	2228 ± 193
(kcal/kg)	29.9 ± 3	28.6 ± 3	27.3 ± 3	25.7 ± 3	34.6 ± 3	29.6 ± 3
CHO (g) (g/kg)	250.5 ± 21.6	252.9 ± 37.4	226.7 ± 31.6	235.1 ± 24.3	300.5 ± 43.9	215.6 ± 30.0
	3.4 ± 0.3	3.5 ± 0.5	3.1 ± 0.4	3.2 ± 0.3	4.1 ± 0.6	3.0 ± 0.4
Fat (g)	89.1 ± 14.3	85.8 ± 7.3	76.7 ± 8.6	93.0 ± 13.2	93.9 ± 10.9	109.0 ± 10.9
(g/kg)	1.2 ± 0.2	1.2 ± 0.1	1.1 ± 0.1	1.3 ± 0.2	1.3 ± 0.2	1.5 ± 0.2
Protein (g)	116.7 ± 10.5	85.8 ± 7.3	96.5 ± 14.4	99.7 ± 15.3	97.8 ± 9.5	100.7 ± 9.7
(g/kg)	1.6 ± 0.1	1.2 ± 0.1	1.3 ± 0.2	1.4 ± 0.2	1.4 ± 0.2	1.4 ± 0.2

Means ± SEM, *n* = 15. EE, total daily energy expenditure; EI, energy intake; CHO, carbohydrate. * Significant difference from EI on corresponding day (*p* < 0.05). ^ No significant difference was observed between the two recovery days. There was a significant difference in TDEE observed between all other days (*p* < 0.05).

**Table 5 nutrients-14-02281-t005:** Average daily energy expenditure and energy intake over 6 different load days for forwards and backs.

Measure	Fitness	Heavy	Mid-Week Recovery	Light	Game	Post-Game Recovery
Forwards TDEE (kcal)	2646 ± 45	3149 ± 52	1842 ± 44	2251 ± 49	2371 ± 51	1883 ± 42
Forwards EI (kcal)	2533 ± 299	2015 ± 259	1947 ± 360	2324 ± 411	2182 ± 351	2317 ± 323
Backs TDEE (kcal)	2311 ± 15	2799 ± 38	1675 ± 8	1996 ± 28	2174 ± 50	1694 ± 31
Backs EI (kcal)	1791 ± 211	2231 ± 268	1881 ± 194	1891 ± 143	2549 ± 268	2125 ± 207

Means ± SEM, *n* = 15, 9 forwards, 6 backs. EE, total daily energy expenditure; EI, energy intake; CHO, carbohydrate. Forwards’ TDEE was significantly different (*p* < 0.01) between all days except for the two recovery days, and light vs. game day. Backs’ TDEE was significantly different (*p* < 0.01) between all days except for the two recovery days, and fitness vs. game day. No significant differences in EI across day types were seen in either position group.

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
