# Peer review of "Dietary Intake over a 7-Day Training and Game Period in Female Varsity Rugby Union Players"

_nutrients, 2022, doi:10.3390/nu14112281_

Round 1

Reviewer 1 Report

There is a paucity of studies on energy balance in team sports and particularly in female athletes. Accordingly, this study has value despite the apparent limitations and the findings are in general agreement with previous studies but could be improved with further consideration.

Major

  1. I’m sure the authors have probably had other reviews highlighting the methodological limitations of the current study, and to their credit acknowledge this. Nonetheless, I think your findings are worthwhile to provide the sport science/dietetics community with additional data for the continual development of dietary recommendations in team sports and across different levels of competition. However, I do think more context is needed to emphasise all three primary measures in the current study are estimates – no quantification of RMR via indirect calorimetry, no device used for understanding physical activity outside of training and match, plus the well-known dietary recording limitations. In fact, it is likely that reported differences in the current study are not greater than error of measurements.
  • In addition, if validity or at least reliability data is available for GPS derived estimates of energy expenditure (I would not be willing to take Catapult’s word for it, and as far as I am aware they have not published/made available how they calculate this for Playertek systems. Perhaps PMID: 25804422 may help?) this should be reported. If it has not, as I expect, it should be stated that the validity and reliability is unknown. So, yes there are "significant" margins of error, and that is not unique to this study, but the outcomes do agree with other studies showing inadequate energy intake in team sports (e.g. PMID: 33900260 ).
  • Also, a case point, text similar to the first line of the abstract that uses the term “quantified”, I would suggest should read estimated.
  1. The discussion is very descriptive without extending the dialogue on findings in comparison to other studies within team sport and/or male participants. It does not interrogate potential contributions or issues as well as it could. Points of discussion from which at least one could be included, and I think would enhance the manuscript substantially:
  • If EI is typically below EE across different team sports (any of soccer, rugby codes, AFL, NHL, NFL, netball, field hockey) and is consistent between them then what could be the common issue(s)? See PMID: 31126159 as a starting point.
  • Are the dietary recommendations accurate or appropriate? There’s potential for endurance exercise bias in the recommendations, I think if there is a common low EA/balance in team sports (male-female, professional-amateur) are the recommendations wrong (for team sport)?
  • Is it only important that EA is good on match day, or maybe also the day before matches, for rugby performance at this level (glycogen is not as limiting/necessary in rugby codes than soccer and AFL, and endurance events)?
  • Were there any sports foods/supplements used by your participants/athletes? How could that impact EI and EA if a strategy was put in place to include a supplement program?
  1. Line 241-44: The analytical approach of adding 10% onto EI is not included in the methods, and therefore also does not have a rationale provided (with appropriate referencing) that provides context for any potential under reporting.

Minor

The first three references in the introduction refer to rugby league. There are very important differences between rugby league and rugby union, so more appropriate references should be included. Here is a good example: Duthie et al. Sports Med 2003 PMID: 14606925.

Line 30: what is optimal? Like the other characteristics listed I think “high levels” is more appropriate.

Line 49: Being pedantic, I’m not sure you can completely exhaust glycogen stores.

Line 54: … muscle building, … (but also) repair and regeneration

Anthropometry measurements: while this is not a primary measure in the study, data are reported relative to total lean body mass, for which BIA can be questionable. Is there validity data available for this equipment, I assume not from your lab but within the literature?

Line 212: I think you’re missing requirements i.e. 6% less than their estimated TDEE requirements.

Line 217-19: I don’t think you mentioned this intra-week effect in the discussion, only week to week, but I think this over-consuming on lighter load days pattern could also contribute to remaining weight stable within weeks, and therefore not necessarily week to week.

Line 257-58: I’m not convinced “hitting” is appropriate nomenclature for this (or any) scientific paper.

Line 290-91: I suggest moving this brief, somewhat random sentence to a more appropriate section within the discussion.

Line 351-353: I think suggesting the effects of short-term LEA can directly hinder muscle performance and increase injury potential is overstated, at best (if at all) it may indirectly impact these factors.

Reviewer 2 Report

Overall Comments:

In the present study, the authors conducted an observational study surrounding parameters of energy availability, requirements, and intake in female rugby players. The findings demonstrated that female rugby players in general, consume fewer calories than they expand and these are in the form of carbohydrates. Overall, I find the paper executed nicely and written well. I do not have any glaring major concerns, although some minor edits will assist this paper in its readability.

Minor Comments:

Line 45: Since you’ve already abbreviated energy intake, do so here to EI and check the paper for any other inconsistencies with abbreviations

Line 52: Change 24+ to ≥24 hours

Line 55-57: Starting with the sentence “Literature…”, please re-word this. I read this sentence several times and am unsure what is being communicated here.

Line 57-61: I’d highly recommend removing this section. You speak of essential fatty acids in a way that makes the reader assume you collected this data in the methods and when the methods appear, it’s clear you did not account for essential fatty acids. Why include this then in the introduction? Keep the introduction narrow and concise, consistent with your methodology.

Line 89: Remove subject and begin the sentence with Characteristics. Also, everywhere else you refer to subjects as participants, so keep this consistent including in your Table 1 headers.

Line 99: Cite your stadiometer and scale

Table 2: Just thought this was nicely displayed and well-done. This is often overlooked by research teams conducting similar type of studies.

Line 202: Change energy availability to EA

Line 204: Italicize post hoc

Lien 234: Please correct the (units?) statement

Line 304: Please remove “other” so it just reads, Few studies…

Line 334: Correct mplayers

Line 350 and 353: There is no reason to abbreviate MPS here since its used only twice. Just spell it out in both areas. There are enough abbreviations at this point to not include MPS as well now.

Limitations: Somewhere in the discussion, there needs to be added a brief but concise statement that a limitation to this study is the fact these rugby players were nutritionally counseled prior to collection of their data. The evidence is clear that athletes, when counseled on nutrition, likely make better nutrition choices which may include being mindful of their caloric intake. While I do not see this limitation as a bad aspect to the study per se, I do think it may have potentially masked even further caloric/CHO deficits. Is there a way to collect this data without nutrition counseling? I would probably say no, but still, I think this is important for the practioner to keep this in mind when reading your paper.

Holtzman B, Ackerman KE. Measurement, Determinants, and Implications of Energy Intake in Athletes. Nutrients. 2019 Mar 19;11(3):665. doi: 10.3390/nu11030665. PMID: 30893893; PMCID: PMC6472042.
